# The Role of Neutrophils in Biomaterial-Based Tissue Repair—Shifting Paradigms

**DOI:** 10.3390/jfb14060327

**Published:** 2023-06-19

**Authors:** Ana Beatriz Sousa, Judite N. Barbosa

**Affiliations:** 1i3S—Instituto de Inovação e Investigação em Saúde, Universidade do Porto, Rua Alfredo Allen, 208, 4200-125 Porto, Portugal; beatriz.noval@gmail.com; 2INEB—Instituto de Engenharia Biomédica, Rua Alfredo Allen, 208, 4200-125 Porto, Portugal; 3ICBAS—Instituto de Ciências Biomédicas Abel Salazar, Universidade do Porto, Rua de Jorge Viterbo Ferreira, 228, 4050-313 Porto, Portugal

**Keywords:** biomaterials, neutrophil, foreign body response, neutrophil polarization, immunomodulatory biomaterials

## Abstract

Tissue engineering and regenerative medicine are pursuing clinical valid solutions to repair and restore function of damaged tissues or organs. This can be achieved in different ways, either by promoting endogenous tissue repair or by using biomaterials or medical devices to replace damaged tissues. The understanding of the interactions of the immune system with biomaterials and how immune cells participate in the process of wound healing are critical for the development of successful solutions. Until recently, it was thought that neutrophils participate only in the initial steps of an acute inflammatory response with the role of eliminating pathogenic agents. However, the appreciation that upon activation the longevity of neutrophils is highly increased and the fact that neutrophils are highly plastic cells and can polarize into different phenotypes led to the discovery of new and important actions of neutrophils. In this review, we focus on the roles of neutrophils in the resolution of the inflammatory response, in biomaterial–tissue integration and in the subsequent tissue repair/regeneration. We also discuss the potential of neutrophils for biomaterial-based immunomodulation.

## 1. Introduction

Neutrophils are polymorphonuclear leukocytes derived from bone marrow hematopoietic stem cells. They are granulocytes, together with eosinophils and basophils, and are a type of myeloid leukocyte that in humans represents about 50–70% of the circulating white blood cells [1]. Neutrophils protect the host against pathogenic agents and are essential to control infections [2]. Neutrophils present a segmented nucleus and a cytoplasm with granules, filled with many different proteins, and vesicles [3]. The segmented or multi-lobular nucleus is a rather interesting feature since it facilitates the movements of these cells through tight gaps between cells and through slight spaces in the extracellular matrix (ECM) [4]. In the cytoplasm, neutrophils have primary, secondary and tertiary granules and secretory vesicles. The primary granules release enzymes such as myeloperoxidase and neutrophil elastase; the secondary granules release molecules such as peptidoglycan recognition protein and lactoferrin; the tertiary granules release matrix-degrading proteins such as matrix metalloproteinase (MMP)-9 [5]. Neutrophils are short-lived cells with an estimated circulation half-life of about 6–9 h. Nevertheless, as these cells become activated, their longevity increases, which assures their presence at inflammatory sites [6]. Neutrophils are the first line of defense and the first immune cells to arrive at the inflammatory site, where they have different functions, as presented in Figure 1 [7,8,9,10,11]:*(i)* Present high capacity for phagocytosis and may degrade foreign bodies. The neutrophil membrane is able to engulf a microorganism or a particle, and phagocytosis occurs within few minutes.*(ii)* Produce reactive oxygen species (ROS) such as superoxide, hydroxyl ion and hydrogen peroxide.*(iii)* Undergo degranulation or exocytosis, releasing highly cytotoxic products such as proteases and anti-microbial molecules.*(iv)* Release neutrophil extracellular traps (NETs) that consist of the extrusion of their DNA in a net-like configuration together with proteases and antimicrobial molecules that will trap and kill pathogens. Neutrophils can sense the size of pathogens and, if they are unable to phagocyte the pathogen due to their size, they will release NETs.

These powerful actions of neutrophils, if persistent, may lead to an uncontrolled inflammatory response or to a chronic response and can impair tissue repair/regeneration, and this may be the reason why neutrophils were not considered for a long time as important players in the context of tissue repair. In fact, neutrophils were seen as detrimental in terms of tissue repair/regeneration. However, new discoveries on neutrophil roles in tissue repair are being disclosed, and the simplistic view on the neutrophil role of eliminating pathogens is being reviewed [12], as we will discuss throughout this review. 

## 2. The Neutrophil in the Foreign Body Response to Biomaterials

Biomaterials will be recognized by the host as foreign and will elicit a foreign body response (FBR). This response can be summarized in four different phases: (i) protein adsorption; (ii) acute inflammation; (iii) chronic inflammation; and (iv) formation of a fibrous capsule. After implantation, blood proteins will adsorb to the surface of the biomaterial, leading to the activation of the coagulation and complement systems that will contribute to the recruitment of inflammatory cells to the implant site. The acute inflammatory response is initiated upon the arrival of inflammatory cells and if not resolved will evolve to a chronic response. Ultimately, fibroblasts will be recruited and activated and will produce collagen, causing the formation of a fibrous capsule around the implant. This collagenous fibrous capsule will not only isolate the implanted material from the surrounding tissues but will also impair its function [13,14,15,16,17]. 

Neutrophils are the first immune cells to arrive to the implant site. They extravasate from blood vessels and migrate to the site of injury in response to inflammatory signals, such as cytokines, leukotrienes (LT) and histamine produced by platelets and endothelial cells and from the injured cells around the implanted biomaterial [18]. Indeed, injured cells have an essential role in the recruitment of inflammatory cells to the implant site through the release of DAMPs (danger-associated molecular patterns), also known as alarmins or danger signals [17,19]. One of the first interleukins (IL) produced in inflammatory sites is IL-8, which acts as a chemoattractant and induces the migration of neutrophils to the implant site. These recruited neutrophils will also produce IL-8, causing the recruitment of further neutrophils. Leukotriene B4 (LTB4) also has an important role in neutrophil recruitment [20]. At the implant site, neutrophils will initiate a phagocytic response; they will secrete proteolytic enzymes and ROS by their cytoplasmic granules, creating a degradative microenvironment that may corrode the biomaterial surface [21,22]. Neutrophils will also release NETs, which have been associated with a fibrotic response and therefore contribute to the formation of a denser fibrous capsule around the implanted biomaterial, which will impair its function [23,24,25]. Neutrophils release NETs when they are unable to phagocyte to potentiate their degradative capacity. Interestingly, this can be compared with the formation of foreign body giant cells (FBGCs) by macrophages upon frustrated phagocytosis with the same aim of enhancing their degradative capacity [5,26]. The formation of NETs on the surface of biomaterials has been associated with the promotion of the inflammatory response and fibrosis [27], although further research is needed to fully understand this mechanism.

Neutrophils will also secrete macrophage inflammatory protein (MIP)-1β and monocyte chemotactic protein (MCP)-1, which act as chemoattractants for other immune cells such as monocytes, macrophages and lymphocytes [28]. Neutrophils are the predominant immune cell type during the first two days after biomaterial implantation, after which they are gradually replaced by monocytes/macrophages. 

## 3. Interactions between Neutrophils and Biomaterials

Normally, neutrophils are cleared from a wound after a few days. However, it was reported that around implanted biomaterials, neutrophils may persist for several weeks [29,30], suggesting a more important role than initially considered. As such, the neutrophil response to biomaterials has gained increased attention. In the studies reported above, neutrophils were identified either by immunohistochemistry or by flow cytometry, using the cell surface protein Ly6G (lymphocyte antigen 6 family member G) as a neutrophil marker.

Jhunjhunwala et al. [29] have investigated the neutrophil response to microcapsules, implanted in the peritoneal cavity, of five different biomaterials: alginate, glass, polystyrene, poly (methyl)-methacrylate and poly (lactic-co-glycolic) acid. They observed a significant increase in neutrophils in the peritoneal exudates after microcapsule implantation, and they observed the formation of some NET-like structures on the surfaces of the microcapsules. Fetz et al. [23] evaluated the neutrophil response to electrospun polydioxanone, collagen type I and blended polydioxanone–collagen templates with different diameters both in vitro and in vivo. Their results revealed that the smaller diameter fibers presented more NETs at the surface, leading to the fibrous encapsulation of the implant, whereas the larger diameter templates had fewer NETs at the surface facilitating implant–tissue integration. Abaricia et al. [31] studied in vitro the role of neutrophils in the response to titanium implants with different surface topography and hydrophilicity. They observed that the rough and hydrophobic implants caused a decrease in enzyme and pro-inflammatory cytokine production together with a decrease in NET formation in comparison with the smooth and the rough implants. The authors conclude that the neutrophil response is influenced by the biomaterial surface characteristics. In another study using polydimethylsiloxane substrates, these authors also concluded that the formation of NETs was dependent of the stiffness of the material. Materials with higher stiffness led to an increase in NET formation and increased secretion of pro-inflammatory cytokines [32]. Avery et al. [33] investigated in vivo the immune response to titanium, titanium alloy, 316L stainless steel and polyetheretherketone implants. They concluded that the chemical composition of the implants affects the neutrophil response. They observed higher numbers of neutrophils with increased production of myeloperoxidase and elastase as well as an increase in the formation of NETs in stainless steel and polyetheretherketone implants when compared with the titanium. Fong et al. [34] analyzed the influence of different functional groups in isodecyl acrylate polymers on neutrophil behavior. They observed that amine groups induced a higher production of pro-inflammatory cytokines and an increased formation of NETs in comparison with carboxyl groups, demonstrating the importance of the chemical characteristics of the implanted material in the neutrophil response. Wesdorp et al. [35] have studied the response of primary human neutrophils to natural and synthetic materials. They observed that in the presence of natural materials, neutrophils had higher survival rates, decreased enzyme and cytokine production and decreased NETs formation in comparison with the synthetic materials. The authors state that it is important to conduct further studies to correlate neutrophil response to tissue repair/regeneration.

From an opposite perspective, Peng et al. [36] proposed an innovative solution for developing orthopedic implants with antimicrobial properties. They have used zinc implants to induce the formation of NETs and therefore prevent possible implant-associated infections. 

In summary, several biomaterial-related factors will influence neutrophil response, such as the type and origin of the biomaterial (natural or synthetic), the surface topography, hydrophobicity/hydrophilicity as well as the chemical composition. Based on the studies herein discussed, it is suggested that hydrophobic materials, materials with lower stiffness and natural materials in comparison with synthetic ones will lead to a more anti-inflammatory response of neutrophils.

In conclusion, tuning the biomaterial physical and/or chemical properties appears as an interesting approach to modulate the neutrophil response and ultimately to facilitate biomaterial–tissue integration.

## 4. Neutrophils in Wound Healing, Tissue Repair and Regeneration

After injury, the wound healing process is activated, commonly leading to a non-functioning mass of fibrotic tissue known as a scar. The recreation of injured tissues without fibrosis or scar formation is considered tissue regeneration. Several strategies can be used to modulate the scar response and potentially enhance tissue regeneration. The cells of the immune system and inflammatory mediators such as cytokines may allow the design of tissue engineering strategies that modulate the healing response in a way favorable to regeneration. One of the major challenges in regenerative medicine is how to optimize tissue regeneration in the body by manipulating its natural ability to form scars upon injury [37].

### 4.1. The Conventional Perspective

In an attempt to understand the role of neutrophils in the process of wound healing, in 1972, Simpson et al. [38] used guinea pigs wherein neutrophils were depleted using anti-neutrophil serum. The wound healing model used comprised linear incisions in the skin of the dorsal area. The authors report that no differences were found between control and neutropenic wounds in terms of tissue repair. These authors have reduced the neutrophil role to the clearance of microorganisms, having an important role against infection and no other role in the process of wound healing. Some years later, in 2003, Dovi et al. [39] reported an investigation conducted in neutrophil-depleted mice using the excisional injury model. Neutrophils were depleted using rabbit anti-mouse neutrophil serum. They observed that in neutrophil-depleted animals, the re-epithelialization of wounds was accelerated. These authors suggest that neutrophils may delay wound closure. More recently, Wong et al. [40], using an excisional skin wound model in diabetic and normoglycemic mice, concluded that the formation of NETs by neutrophils delayed wound healing, particularly in diabetic mice. Although the above-mentioned studies suggest a negative role of neutrophils in wound healing, Devalaraja et al. [41] reported a positive effect. They have investigated the role of CXC chemokines in wound healing using an excisional wound model in CXC chemokine receptor-2 (CXCR2) wild-type and knockout mice. They observed that the knockout mice presented defective recruitment of neutrophils and delayed wound healing. These authors suggest that wound healing could be in part mediated by neutrophil recruitment or that the initial neutrophil recruitment would influence the recruitment of further immune cells. This study may have contributed to starting the change in perspective of the role of neutrophils in wound healing. 

### 4.2. The Paradigm Shift

Since it was realized that upon activation, the longevity of neutrophils increases significantly, it was suggested that their role in the inflammatory and wound healing processes could be more relevant than previously considered [20]. Neutrophils are presently viewed as multifaceted cells that participate in different and specialized functions that we will discuss in the following sections.

#### 4.2.1. Neutrophil Polarization

Neutrophils have been considered for a long time as a homogeneous type of cell. However, recent advances in neutrophil biology have led to the perception that neutrophils are in fact a pool of heterogeneous cells. The microenvironment will influence the function and phenotype of these cells. Neutrophils can have positive and negative actions for the host, most likely depending on the stimuli that they receive [42]. Neutrophils are highly plastic cells that can polarize in response to different cues into different phenotypes. Although the classification of neutrophil subpopulations is still a controversial topic that needs to be normalized, neutrophils polarize into N1 and N2 phenotypes (Figure 2), as in the case of M1 and M2 macrophages. Fridlender et al. [43,44] were the first to describe these different neutrophil subtypes. The N1 pro-inflammatory neutrophils are anti-tumoral and the N2 anti-inflammatory are pro-tumoral neutrophils. The N1 subpopulation consists of pro-inflammatory neutrophils that activate the immune system and present higher expression of cytokines, decreased levels of arginase and an increased capacity to kill tumor cells. The N2 subpopulation consists of anti-inflammatory neutrophils characterized by the expression of increased levels of arginase, MMP-9 and vascular endothelial growth factor (VEGF). In terms of morphology, N1 neutrophils present a hyper-segmented nucleus, whereas N2 neutrophils present an immature nucleus in a ring-like shape [5,43,45,46]. The nucleus of a cell is its stiffest and largest component, therefore having an impact on cell migration. The lobular shape of the neutrophil facilitates its motility. The multi-lobular nuclei of the N1 pro-inflammatory neutrophils will allow a faster migration through the endothelium and through the ECM [47]. In an animal model of myocardial infarction, different subpopulations of neutrophils were identified. After injury, N1 pro-inflammatory neutrophils were detected, whereas N2 anti-inflammatory neutrophils appeared later, at about day 5 after injury. The N2 subpopulation expressed the neutrophil marker Ly6G and the mannose receptor CD206. The N1 subpopulation expressed Ly6G and did not express CD206. The authors suggest the possible existence of neutrophils that are specialized in tissue repair [48]. To conclude, it is important to refer to the fact that no definitive surface markers have yet been identified to distinguish N1 and N2 clearly [49] and that the N1/N2 nomenclature is most likely an oversimplification of the different neutrophil subtypes [50].

#### 4.2.2. The Neutrophil in the Resolution of the Inflammatory Response

Interestingly, neutrophils play an important role in the resolution of the inflammatory response through three different mechanisms: *(i)* biosynthesis of specialized pro-resolving lipid mediators (SPMs); *(ii)* efferocytosis, a process through which macrophages clear apoptotic neutrophils; and *(iii)* reverse endothelial migration, when neutrophils migrate back into the vasculature. 

In the final steps of the acute inflammatory response, neutrophils change the biosynthesis of eicosanoids from LTB4 to lipoxin A4, one of the lipidic molecules that belong to the SPMs family. Neutrophils will also contribute to the production of other SPMs such as resolvins and protectin D1 [45]. These SPMs will have important anti-inflammatory and pro-resolutive effects such as inhibiting further neutrophil recruitment, modulating the polarization of immune cells and regulating the elimination of apoptotic immune cells by macrophages through efferocytosis [45,51,52,53]. For a successful resolution of the inflammatory response, there must be a decrease in the number of neutrophils present in the wounded area. Neutrophils leave the implant site by two different mechanisms. Firstly, by efferocytosis, the process of phagocytosis of apoptotic neutrophils by macrophages. Efferocytosis impairs the release of tissue-degrading enzymes and pro-inflammatory mediators and has important anti-inflammatory effects because transforming growth factor (TGF)-β and IL-10 are released, leading to a shift to an anti-inflammatory and pro-regenerative microenvironment, causing M2 macrophage polarization and promoting tissue repair/regeneration [54,55,56]. Neutrophils can also leave the injured site by reverse endothelial migration, which consists of the movement of neutrophils away from injured tissues and back into the vasculature. Neutrophils are able to return to the main circulation instead of undergoing apoptosis. The mechanisms that control neutrophil reverse migration are not clear, however, neutrophils that undergo reverse migration present high expression of intercellular adhesion molecule 1 (ICAM1) and low expression of CXC-chemokine receptor 1 (CXCR1) [20]. It was also suggested that SPMs might be involved in neutrophil reverse migration [20,57].

#### 4.2.3. The Role of Neutrophils in Tissue Repair/Regeneration

The perception of the role of neutrophils in tissue repair/regeneration has been changing over the last years. Both positive and negative effects are attributed to these cells. Neutrophils release ROS and through degranulation they release toxic mediators, which will not only cause further tissue injury but will also induce more neutrophil recruitment. Neutrophils also release MMPs such as MMP-9 that will cleave the ECM that again may further stimulate immune cells and will cause more local tissue injury [58,59]. Additionally, an excessive release of NETs may lead to the formation of a dense fibrotic capsule and may impair biomaterial–tissue integration [25,26]. On the other hand, important functions in the wound healing process are attributed to neutrophils, which are summarized in Figure 3. Lately, a positive role for neutrophils in the context of tissue repair has been reported in terms of anti-inflammatory and pro-regenerative actions [60,61,62]. Neutrophils upregulate the expression of important cytokines and chemokines such as tumor necrosis factor (TNF)-α, IL-1β, IL-6 and MCP-1 that will recruit other immune cells important for the wound healing process [63,64]. Neutrophils have an important role in promoting angiogenesis through an increased expression of VEGF, fibroblast growth factor (FGF) and TGF-β. Neutrophils influence the activation and proliferation of fibroblasts and keratinocytes by the secretion of IL-8, IL-1β and MCP-1 [55,64,65]. These cells also affect tissue remodeling via release of urokinase-type plasminogen activator (uPA) [66]. Neutrophils also have the important role of cleaning the debris and necrotic tissue at the injury site [56]. They release elastase, heparanase and cathepsin that will clear tissue debris and damaged ECM [67].

Neutrophil plasticity and heterogeneity will most likely contribute to the different type of response in terms of tissue repair/regeneration [48]. It is likely that an N1 pro-inflammatory phenotype will cause deleterious effects, whereas an N2 anti-inflammatory phenotype will have beneficial effects in terms of tissue repair.

#### 4.2.4. The Neutrophil in Biomaterial-Based Immunomodulation

Regenerative medicine aims to restore, repair or replace damaged or diseased cells, organs and tissues. This can be achieved using, for example, gene therapy, soluble mediators and stem cell transplantation. Biomaterials appear as a powerful tool for the reconstruction and regeneration of damaged tissues. They have a crucial role because they represent an alternative to conventional implantation or replacement of organs and tissues. The application of biomaterials presents several promising possibilities, such as the chance of combining biomaterials with cells, antibiotics, growth factors and chemotherapeutic agents. 

The reactions at the interface between cells and the surface of the biomaterial have a key role in a successful tissue repair and regeneration. The acute inflammatory response triggered after implantation will favor tissue regeneration, nevertheless, the persistence of the inflammatory response will impair healing [68,69,70,71].

The successful application of biomaterials in tissue regeneration requires the modulation and resolution of the inflammatory response. The modulation of the neutrophil response arises as a new strategy for the development of novel materials capable of accelerating tissue repair/regeneration.

Li et al. [72] investigated the importance of the osteoimmune microenvironment in bone repair after the implantation of different biomaterials in murine tibia. They suggest that neutrophils may have a positive effect in bone formation possibly through the recruitment of bone marrow stromal cells via the chemokine (C-X-C mode) ligand (CXCL)12/CXC chemokine receptor (CXCR)3 axis. Gao et al. [73] have developed a neutrophil immunomodulatory biomaterial through the incorporation of growth factors secreted by neutrophils into a gelatin methacrylate hydrogel. The authors differentiate in vitro N1 neutrophils using lipopolysaccharide (LPS) and N2 neutrophils using TGF-β. Afterwards, they incorporated into the hydrogel conditioned media of N2 neutrophils and in an in vivo subcutaneous model observed a reduction in the recruitment of inflammatory cells and increased angiogenesis. Li et al. [74] have explored the immunomodulatory properties of strontium, namely its effects on neutrophils. They have produced gelatin scaffolds with incorporated strontium-hydroxyapatite, gelation scaffolds with hydroxyapatite and pure gelatin scaffolds. The strontium-hydroxyapatite scaffolds induced more neutrophil polarization towards N2 neutrophils and promoted angiogenesis both in vitro and in vivo when compared with the other scaffolds. The authors conclude their report by suggesting that neutrophil-based immunomodulatory biomaterials are a promising approach for tissue engineering applications.

In conclusion, the control of the N1/N2 neutrophil ratio can possibly be as important as the control of the M1/M2 macrophage ratio and can be a promising tool to facilitate biomaterial–tissue integration [5]. The studies conducted so far are quite scarce; however, the results obtained suggest that the neutrophil could be in fact a rather promising target for biomaterial-based immunomodulation. Nevertheless, the modulation of neutrophil phenotype may have side effects such as increased risk of infection that need to be carefully taken into consideration [75].

## 5. Conclusions and Future Perspectives

The role of neutrophils in the inflammatory response has gained increased interest over the last years since it was realized that neutrophils are more than short-lived phagocytic immune cells. Conventionally, neutrophils have been considered as having harmful effects in terms of tissue repair and regeneration. Nevertheless, new studies revealed a positive role for neutrophils during the resolution of the inflammatory response and the subsequent tissue repair. Taking into consideration the herein described roles of neutrophils in the resolution of the acute inflammatory response, in facilitating biomaterial–tissue integration and ultimately in tissue repair/regeneration, we suggest that the development of new immunomodulatory biomaterials should focus on the modulation of the neutrophil response to achieve an ideal biomaterial–tissue integration. Actually, an accurate modulation of the neutrophil response will influence the subsequent response of other important immune cells such as the macrophage. 

The understanding that neutrophils are rather plastic cells and can polarize into different effector phenotypes opens exciting and new possibilities in terms of the development of new immunomodulatory biomaterials. So far, the results reported in the literature that focused on the neutrophil for biomaterial-based immunomodulation are rather promising, although there are very few studies available. The potential for neutrophil-based immunomodulation is, in our opinion, high, but several aspects need to be addressed for a better understanding of the potential of these cells and how to modulate them:(i)Neutrophil polarization into an N1 or N2 phenotype has been more studied in cancer models. It would be important to study and characterize these phenotypes in a wound healing environment.(ii)Taking into consideration the interesting mechanism of neutrophil reverse endothelial migration, it would be interesting to explore the possibility of, under a chronic inflammatory microenvironment, inducing the reverse migration of N1 neutrophils while maintaining the N2 neutrophils at the inflammatory site.(iii)The macrophage polarization into M1 and M2 has been extensively investigated, and it is known that an increased M2/M1 ratio is favorable for tissue repair. It is necessary to understand the relationship between neutrophil and macrophage polarization. Will neutrophil polarization influence macrophage polarization and the other way around also?(iv)Among the N1 and N2 phenotypes, are there different subsets as in the case of macrophages? Which markers can be used to identify different neutrophil subsets? It is critical to find adequate antibodies for an accurate identification of these cell subtypes.

The recent advances in neutrophil biology have opened many exciting and new possibilities for novel research lines. It is necessary to re-evaluate neutrophil involvement in the biological response to biomaterials. It is also necessary to broaden the design of new immunomodulatory materials through the modulation of the neutrophils towards an N2 anti-inflammatory and pro-regenerative phenotype.

With this review, we hope to have convinced the readers of the key roles of neutrophils in the inflammatory response, in its resolution and in the ensuing tissue repair and to look at the neutrophil as a crucial player in biomaterial–host interactions.

Further research work is needed to fully clarify the anti-inflammatory and pro-resolutive neutrophil functions, to understand how neutrophils can accelerate tissue repair/regeneration and finally to successfully translate it into clinical applications.

## Figures and Tables

**Figure 1 jfb-14-00327-f001:**
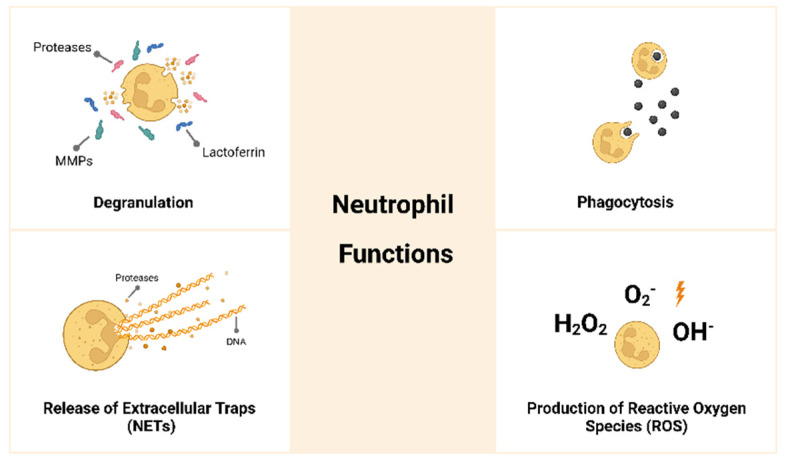
The role of neutrophils in the acute inflammatory response. Neutrophils are the first immune cells to arrive at the inflammatory site. Neutrophils are highly phagocytic cells; they produce reactive oxygen species (ROS) and undergo degranulation, releasing highly toxic products, creating a degradative microenvironment. They also release neutrophil extracellular traps (NETs) to trap and kill pathogenic agents.

**Figure 2 jfb-14-00327-f002:**
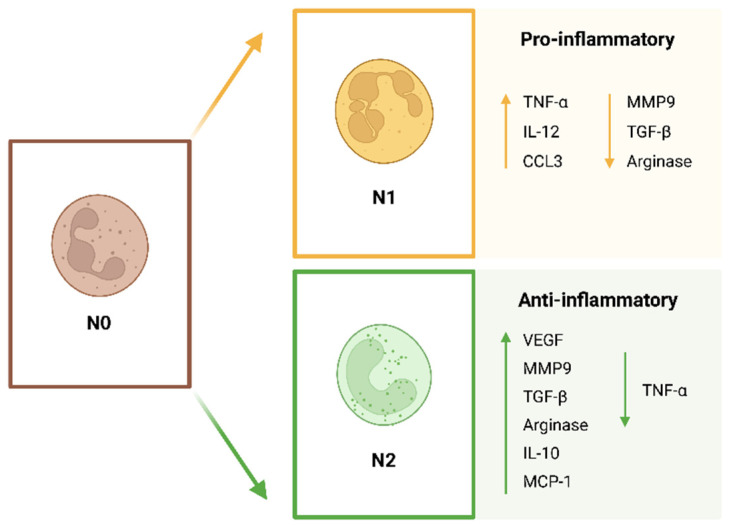
Neutrophil polarization. Neutrophils (N0: non-activated) are plastic cells that can undergo polarization into an N1 pro-inflammatory or N2 anti-inflammatory phenotype. (TNF: tumor necrosis factor; IL: interleukin; CXCL: CXC chemokine ligand; MCP: monocyte chemotactic protein; MMP: matrix metalloproteinase; TGF: transforming growth factor; VEGF: vascular endothelial growth factor).

**Figure 3 jfb-14-00327-f003:**
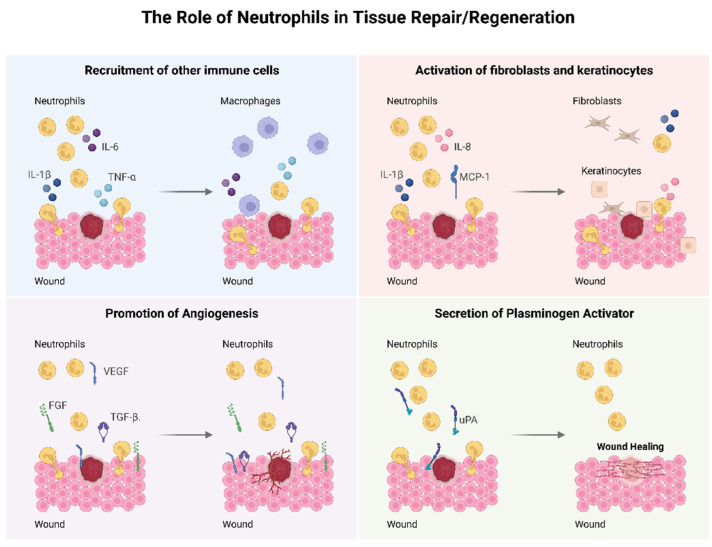
The neutrophil in tissue repair. Neutrophils upregulate the expression of several cytokines and chemokines that will lead to the recruitment of other immune cells, promote angiogenesis through the release of growth factors, activate fibroblasts and keratinocytes by cytokine production and influence tissue repair via secretion of uPA. (IL: interleukin; TNF: tumor necrosis factor; MCP: monocyte chemotactic protein; VEGF: vascular endothelial growth factor; FGF: fibroblast growth factor; TGF: transforming growth factor; uPA: urokinase-type plasminogen activator).

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
