# Peer review of "The Role of Neutrophils in Biomaterial-Based Tissue Repair—Shifting Paradigms"

_jfb, 2023, doi:10.3390/jfb14060327_

Round 1
Reviewer 1 Report
1. Generally, the topic is relevant in the field but not totally original in the field, neither address a specific gap in the field.
2. Please add a highlight of “biomaterial-based tissue repair”, its characteristics different from other type of tissue repair and the importance of neutrophils on biomaterial-based tissue repair.
3. What does current review add to the subject area compared with other published material? For example, more details or summary in the reaction of neutrophils on different type biomaterial could be interesting to the readers, such as synthetic or biological or metal etc.,.
4. Specific improvements in methodology and results from different studies, the brief description in how identification of neutrophils is suggested. Did neutrophil count quantify or qualify?
5. Neutrophil represents innate immune response and the most detrimental reaction is adaptive immune response, such as cytotoxic effect by T cell, which may cause chronic rejection. Is there any connection of the role of neutrophil with adaptive immune response in biomaterial-based tissue regeneration?
6. Adding more important and key references about the role of inflammatory and immune reaction about biomaterial-based tissue repair is suggested.
7. Potential side effect of neutrophil modulation on human body is worthy of discussion.
8. In conclusion, the harmful effects of neutrophils on wound healing is not true.
Minor editing of English is suggested
Author Response
REVIEWER # 1
- Generally, the topic is relevant in the field but not totally original in the field, neither address a specific gap in the field.
The idea behind this manuscript was to disclosure the potential of neutrophils in biomaterial-based tissue repair and in the development of new immunomodulatory biomaterials. There has been a clear change in the perception of the importance of neutrophils in the foreign body response to biomaterials, in the resolution of the inflammatory response and ultimately in tissue regeneration. There are scarce studies available in the literature that explore the neutrophil in these different topics and we found this a rather interesting area with a huge potential for the development of new biomaterial-based approaches for regenerative medicine.
- Please add a highlight of “biomaterial-based tissue repair”, its characteristics different from other type of tissue repair and the importance of neutrophils on biomaterial-based tissue repair.
Regarding the “biomaterial-based tissue repair”, we have included the following paragraph in the manuscript:
“Regenerative medicine aims to restore, repair or replace damaged or diseased cells, organs and tissues. This can be achieved using for example gene therapy, soluble mediators, stem cell transplantation. Biomaterials appear as a powerful tool for the reconstruction and regeneration of damaged tissues. They have a crucial role because they represent an alternative to conventional implantation or replacement of organs and tissues. The application of biomaterials presents several promising possibilities as the chance of combining biomaterials with cells, antibiotics, growth factors and chemotherapeutic agents.”
Regarding the importance of neutrophils on biomaterial-based tissue repair, in section 4.2.4 we discuss the importance of neutrophils in biomaterial-based immunomodulation for tissue repair. We conclude that there is a lot of potential in this area that needs to be deeply explored. The reports in the literature on biomaterial-based immunomodulation and biomaterial-based tissue repair are mainly focused on the macrophage and very few explore the neutrophil. This paradigm shift around the role of neutrophils opens many new possibilities for the development of biomaterials for tissue repair.
- What does current review add to the subject area compared with other published material? For example, more details or summary in the reaction of neutrophils on different type biomaterial could be interesting to the readers, such as synthetic or biological or metal etc.,.
The purpose of this review is to call the reader’s attention for the yet underexplored role of neutrophils in biomaterial-based tissue repair. Taking into consideration the acknowledgment on the increased longevity of neutrophils upon activation, and the importance of neutrophils in the resolution of the inflammatory response and in tissue repair we can conclude that neutrophils, although to date they have not been much explored, should be considered as rather interesting target cells. In our opinion, this manuscript opens new avenues for immunomodulation in tissue repair based on the neutrophil response.
On Section 3: “Interactions between Neutrophils and Biomaterials” we discuss that several properties of the biomaterials will influence neutrophil response such as stiffness, hydrophobicity / hydrophilicity, chemical composition, among other examples. To improve this section we have added more information:
“In another study using polydimethylsiloxane substrates, these authors also concluded that the formation of NETs was dependent of the stiffness of the material. Materials with higher stiffness lead to an increase in NET formation and increased secretion of pro-inflammatory cytokines [32].”
We have also add the following paragraph in the end of the section:
“In summary, several biomaterial-related factors will influence neutrophil response such as the type and origin of the biomaterial (natural or synthetic), the surface topogra-phy, hydrophobicity / hydrophilicity as well as the chemical composition. Based on the studies herein discussed it is suggested that hydrophobic materials, materials with lower stiffness and natural materials in comparison with synthetic ones will lead to a more anti-inflammatory response of neutrophils.”
- Specific improvements in methodology and results from different studies, the brief description in how identification of neutrophils is suggested. Did neutrophil count quantify or qualify?
In the studies that we discuss throughout this review, neutrophils were indentified either by immunohistochemistry (qualitative technique) or by flow cytometry (quantitative technique). The majority of the reported studies have used flow cytometry analysis to identify neutrophils based on the cell surface protein Ly6G (lymphocyte antigen 6 family member G). To clarify this, we have added the following sentence in the new version of the manuscript:
“In the studies reported above, neutrophils were identified either by immunohistochemistry or by flow cytometry. The majority of the reported studies have used flow cytometry analysis to identify neutrophils based on the cell surface protein Ly6G (lymphocyte antigen 6 family member G).”
- Neutrophil represents innate immune response and the most detrimental reaction is adaptive immune response, such as cytotoxic effect by T cell, which may cause chronic rejection. Is there any connection of the role of neutrophil with adaptive immune response in biomaterial-based tissue regeneration?
This is in fact a very pertinent comment. Interestingly, it has been demonstrated that neutrophils also interact with the cellular components of the adaptive immune system in a bidirectional manner. Neutrophils activate T cells through direct antigen presentation resulting in T cell activation and differentiation both in vitro and in vivo. In turn, T cells secrete factors that promote neutrophil survival and indirectly induce production of neutrophil growth and chemotactic factors. However, there is lack of information regarding the relation between neutrophils with the adaptive immune response in the context of biomaterial-based tissue regeneration. It is in our opinion a subject rather interesting to be studied and explored.
- Adding more important and key references about the role of inflammatory and immune reaction about biomaterial-based tissue repair is suggested.
We thank the reviewer for this comment. In the new version of the manuscript we have included the following sentence and the following references:
“The reactions at the interface between cells and the surface of the biomaterial have a key role in a successful tissue repair and regeneration. The acute inflammatory response trig-gered after implantation will favor tissue regeneration, nevertheless, the persistence of the inflammatory response will impair healing [68-71].”
[68] F. Batool, H. Ozcelik, C. Stutz, P.Y. Gegout, N. Benkirane-Jessel, C. Petit, O. Huck, Modulation of immune-inflammatory responses through surface modifications of biomaterials to promote bone healing and regeneration, J Tissue Eng 12 (2021) 20417314211041428.
[69] K.E. Martin, A.J. Garcia, Macrophage phenotypes in tissue repair and the foreign body response: Implications for biomaterial-based regenerative medicine strategies, Acta biomaterialia 133 (2021) 4-16.
[70] K. Dixit, H. Bora, J. Lakshmi Parimi, G. Mukherjee, S. Dhara, Biomaterial mediated immunomodulation: An interplay of material environment interaction for ameliorating wound regeneration, J Biomater Appl 37(9) (2023) 1509-1528.
[71] R. Mata, Y. Yao, W. Cao, J. Ding, T. Zhou, Z. Zhai, C. Gao, The Dynamic Inflammatory Tissue Microenvironment: Signality and Disease Therapy by Biomaterials, Research (Wash D C) 2021 (2021) 4189516.
- Potential side effect of neutrophil modulation on human body is worthy of discussion.
We thank the reviewer for pointing out this aspect. It is in fact a very pertinent question. Considering this comment, we have included in the manuscript the following information:
“Nevertheless, the modulation of neutrophil phenotype may have side effects such as increased risk of infection that need to be carefully taken into consideration [75].”
- Bartneck, J. Wang, Therapeutic Targeting of Neutrophil Granulocytes in Inflammatory Liver Disease, Frontiers in immunology 10 (2019) 2257.
- In conclusion, the harmful effects of neutrophils on wound healing is not true.
In fact, in section 4.2.3 we discuss the effect of neutrophils in tissue repair and both positive and negative effects are attributed to this cells. Probably a N1 pro-inflammatory phenotype will cause deleterious effects whereas a N2 anti-inflammatory phenotype will have beneficial effects.
To clarify this aspect we have added the following sentence to the new version of the manuscript:
“It is likely that a N1 pro-inflammatory phenotype will cause deleterious effects whereas a N2 anti-inflammatory phenotype will have beneficial effects in terms of tissue repair.”
The authors are requested to do minor formatting to correct grammatical errors.
We thank the reviewer for the suggestion and we have asked for the help of a native English-speaking colleague, to help us improve the quality of the manuscript.

Reviewer 2 Report
The manuscript of Sousa and Barbosa describes the role of neutrophils in biomaterial-based tissue repair. In particular, the interaction of the immune system with biomaterials and how these immune cells are involved in the inflammatory response, in tissue regeneration and in immunomodulation are discussed.
Comments:
Line 168 Could the authors discuss the link between the processes of wound healing and tissue regeneration, especially in relation to immune cells and neutrophils?
Figure 2: all the acronyms used in the figure should be specified as full name in the legend of the figure.
Line 235, 242: could the authors give more explanations on what efferocytosis is? I would prefer a brief description at first mention and a more detailed explanation afterwards.
Line 235 May the authors describe better the process of reverse migration and the mechanism through which the movement of neutrophils is regulated?
Line 244 typo By should not be written in capital letters.
Author Response
REVIEWER # 2
Line 168 Could the authors discuss the link between the processes of wound healing and tissue regeneration, especially in relation to immune cells and neutrophils?
We thank the reviewer for this pertinent comment. In fact, we have not clarify the difference between wound healing and tissue regeneration. To clarify this aspect we have made some changes through section 4. We have changed the title of section 4 to: “Neutrophils in Wound Healing, Tissue repair and Regeneration. We have also added the following paragraph:
“After injury, the wound healing process is activated commonly leading to a non-functioning mass of fibrotic tissue known as a scar. The recreation of injured tissues without fibrosis or scar formation is considered tissue regeneration. Several strategies can be used to modulate the scar response and potentially enhance tissue regeneration. The cells of the immune system and inflammatory mediators as cytokines may allow the design of tissue engineering strategies that modulate the healing response in a way favorable to regeneration. One of the major challenges in regenerative medicine is how to optimize tissue regeneration in the body by manipulating its natural ability to form scar upon injury.”
In section 4.2.3 the role of neutrophils in tissue repair is discussed.
Figure 2: all the acronyms used in the figure should be specified as full name in the legend of the figure.
We thank the reviewer for the criticism. We have added the missing acronyms in the legend of figure 2.
Line 235, 242: could the authors give more explanations on what efferocytosis is? I would prefer a brief description at first mention and a more detailed explanation afterwards.
Line 235 May the authors describe better the process of reverse migration and the mechanism through which the movement of neutrophils is regulated?
We thank the reviewer for these two comments and to address them in the new version of the manuscript, we have rearranged section 4.2.2 and add more information as follows:
“4.2.2. The Neutrophil in the Resolution of the Inflammatory Response
Interestingly, neutrophils play an important role in the resolution of the inflammatory response through three different mechanisms: (i) Biosynthesis of specialized pro-resolving lipid mediators (SPMs); (ii) Efferocytosis, a process through which macrophages clear apoptotic neutrophils, and (iii) Reverse endothelial migration, when neutrophils migrate back into the vasculature.
In the final steps of the acute inflammatory response, neutrophils change the biosynthesis of eicosanoids from LTB4 to lipoxin A4, one of the lipidic molecules that be-long to the SPMs family. Neutrophils will also contribute to the production of other SPMs as resolvins and protectin D1 [44]. These SPMs will have important anti-inflammatory and pro-resolutive effects such as inhibiting further neutrophil recruitment, modulating the polarization of immune cells and regulating the elimination of apoptotic immune cells by macrophages through in a process called efferocytosis [44, 50-52]. For a successful resolution of the inflammatory response, there must be a decrease in the number of neutrophils present in the wounded area. Neutrophils leave the implant site by two different mechanisms: by efferocytosis, the process of phagocytosis of apoptotic neutrophils by macrophages. Efferocytosis impairs the release of tissue-degrading enzymes and pro-inflammatory mediators and that has important anti-inflammatory effects because transforming growth factor (TGF)-β and IL-10 are released leading to a shift to an anti-inflammatory and pro-regenerative microenvironment, causing M2 macrophage polarization and promoting tissue repair/regeneration [53-55]. Neutrophils can also leave the injured site by reverse endothelial migration, which consists in the movement of neutrophils away from injured tissues and back into the vasculature. Neutrophils are able to return to the main circulation instead of undergoing apoptosis. The mechanisms that control neutrophil reverse migration are not clear, however neutrophils that undergo reverse migration present high expression of intercellular adhesion molecule 1 (ICAM1) and low expression of CXC-chemokine receptor 1 (CXCR1) [20]. It was also suggested that SPMs might be involved in neutrophil reverse migration [20, 56].”
Line 244 typo By should not be written in capital letters.
We thank the reviewer for pointing out this mistake. We have corrected it in the new version of the manuscript.

Round 2
Reviewer 2 Report
The revised manuscript has been improved.
Minor comment:
Lines 125-126 Could the authors please rewrite the sentence in a different way as it does not seem clear to this reviewer?
Author Response
We thank the reviewer for the comment and we have changed the sentence as follows:
"In the studies reported above, neutrophils were identified either by immunohistochemistry or by flow cytometry, using the cell surface protein Ly6G (lymphocyte antigen 6 family member G) as neutrophil marker".